# Comparison of Lateral Flow Immunochromatography and Phenotypic Assays to PCR for the Detection of Carbapenemase-Producing Gram-Negative Bacteria, a Multicenter Experience in Mexico

**DOI:** 10.3390/antibiotics12010096

**Published:** 2023-01-06

**Authors:** Braulio Josue Mendez-Sotelo, Luis Esaú López-Jácome, Claudia A. Colín-Castro, Melissa Hernández-Durán, Maria Guadalupe Martínez-Zavaleta, Frida Rivera-Buendía, Consuelo Velázquez-Acosta, Ana Patricia Rodríguez-Zulueta, Maria del Rayo Morfín-Otero, Rafael Franco-Cendejas

**Affiliations:** 1División de Infectología, Instituto Nacional de Rehabilitación Luis Guillermo Ibarra Ibarra, Mexico City 14389, Mexico; 2Oficina de Apoyo Sistemático para la Investigación Superior, Subdirección de Investigación Clínica, Instituto Nacional de Cardiología, Mexico City 14080, Mexico; 3Laboratorio de Microbiología, Instituto Nacional de Cancerología, Mexico City 14080, Mexico; 4Infectología, Hospital General Manuel Gea González, Mexico City 14080, Mexico; 5Infectología, Hospital Civil de Guadalajara Fray Antonio Alcalde, Universidad de Guadalajara, Guadalajara 44280, Mexico; 6Biomedical Research Subdirection, Instituto Nacional de Rehabilitación Luis Guillermo Ibarra Ibarra, Ciudad de México 14389, Mexico

**Keywords:** NG-Test CARBA 5^®^, antimicrobial resistance, metallo ß-lactamase, Enterobacterales, *Pseudomonas aeruginosa*

## Abstract

The identification of carbapenemase-producing Enterobacterales and *Pseudomonas aeruginosa* is important for treating and controlling hospital infections. The recommended methods for their identification require a long waiting time, technical training, and expertise. Lateral flow immunoassays such as NG-Test CARBA 5^®^ overcome these needs. We analyzed 84 clinical isolates of carbapenem-resistant Enterobacterales and *P. aeruginosa* from four different hospitals in a two-year period. Antimicrobial resistance patterns were confirmed with the broth dilution method. Evaluation of KPC, VIM, NDM, IMP, and OXA-48-like enzymes was performed and compared to NG-Test CARBA 5 and phenotypic assays. Enterobacterales represented 69% of isolates and *P. aeruginosa* represented 31%. Carbapenemase-producing strains were 51 (88%) of Enterobacterales and 23 (88.4%) of *P. aeruginosa*; 20 (34%) and 23 (88%) were Class B ß-lactamases, respectively. The NG-Test CARBA 5^®^ assay for Enterobacterales showed high sensitivity (98%), specificity (100%), and PPV (100%); however, it did not for *P. aeruginosa*. The Kappa concordance coefficient was 0.92 for Enterobacterales and 0.52 for *P. aeruginosa*. NG-Test CARBA 5^®^ is a fast and easy-to-use assay. In Enterobacterales, we found excellent agreement in our comparison with molecular tests. Despite the low agreement in *P. aeruginosa*, we suggest that this test could be used as a complementary tool.

## 1. Introduction

Hospital-acquired infections, predominantly blood-stream infections and pneumonia, are an important, expensive, and alarming problem for health systems worldwide, as well as being the main cost of bed-days lost [1]. Enterobacterales and *Pseudomonas aeruginosa* are the leading bacteria associated with healthcare infections; furthermore, there has been an alarming increase in multidrug-resistant microorganisms, including resistance to carbapenems [2]; by far the most important mechanism is related to ß-lactamases synthesis, such as carbapenemases [3,4,5].

In a global surveillance study, the prevalence of meropenem non-susceptible Enterobacterales clinical isolates in Latin America was 5.3% vs. 1.4% in the United States of America [6]. Currently, the prevalence of carbapenem resistance in Mexico has been reported from 3 to 12% for Enterobacterales and 35% in *P. aeruginosa*, showing a significant increase in the last decade [7,8]. There are substantial epidemiological and clinical reasons to identify carbapenemases, such as stewardship programs, reduced resistance propagation, and the use of appropriate treatment, while adequately employing new ß-lactamase inhibitors, cefiderocol, or polymyxins. The shortened time-to-result is critical in life-threatening infections such as sepsis [9].

The phenotypic and molecular typification of carbapenemases represents a challenge within clinical microbiology laboratories [10]. The Clinical & Laboratory Standards Institute (CLSI) has published recommendations to make an intentional search for carbapenemases while using phenotypic tests, such as the Carba NP test (colorimetric assay), and modified carbapenem inactivation method (mCIM), with and without EDTA as a quelant to identify metallo ß-lactamases (MBLs), and molecular methods for genes that codify these enzymes [11]. Recently, MALDI-ToF/MS has been evaluated to document carbapenemases, showing faster and more accurate results in comparison to mCIM or Carba NP; however, it is an expensive tool requiring experienced personnel, challenging its worldwide access in routine laboratories, especially in developing countries [12].

Furthermore, some immuno-assays have been widely used in the clinical microbiology field. Lateral flow assay, an immunochromatographic method, has been developed in order to have a fast and straightforward way to identify carbapenemases [13]. NG-Test CARBA-5^®^ (NG-Biotech, Guipry, France) was approved by the FDA in 2019, and it can identify the five most prevalent carbapenemases (KPC, NDM, IMP, VIM, and OXA-48 like), giving results in fewer than 15 min without specialized training. Some previous studies showed 100% sensitivity and 99.75% specificity for Enterobacterales using molecular biology as the gold standard [14]. The most recent works showed 100% positive agreement (CI 95%: 97.8–100%) and 95.1% of negative agreement (CI 95%: 90.3–97.6%) compared to the reference method (mCIM and molecular methods) [14,15]. Although it has exemplary diagnostic performance for identifying Enterobacterales, results for *P. aeruginosa* have unfortunately not been comparable, mainly with IMP producers [16]. According to previous reports of carbapenemases in Mexico, the main enzymes in *P. aeruginosa* and Enterobacterales belong to MBLs [17]. The aim of this study was to evaluate the performance of NG-Test CARBA 5^®^ in clinical isolates in a region where carbapenem resistance due to carbapenemases, such as MBLs, is increasing.

## 2. Results

We analyzed 84 non-duplicated carbapenem-resistant clinical isolates. A total of 58 (69%) strains belonged to Enterobacterales and 26 (31%) belonged to *P. aeruginosa*. The carbapenem resistance distribution for Enterobacterales is shown in Figure 1; for *P. aeruginosa* all strains showed a MIC > 16 μg/mL for meropenem, imipenem, and doripenem. For Enterobacterales, the most frequent bacteria were *Klebsiella pneumoniae*, n = 25 (43%), followed by *Escherichia coli*, n = 18 (31%); the rest of the isolates are shown in Appendix A.

The accuracy of mCIM in Enterobacterales compared to polymerase chain reaction (PCR) was 100% for sensitivity, specificity, NPV, and PPV; the results for the mCIM/eCIM composite and the CarbaNP and CarbaNP/Carba NP-EDTA composite are shown in Table 1. For *P. aeruginosa*, the accuracy of mCIM compared to PCR was 100% in sensitivity, specificity, NPV, and PPV (Table 1).

Carbapenemase production was detected in 51 (88%) of Enterobacterales: 10 were KPC, 17 were NDM, three were VIM, 19 were OXA-48 like, and two were co-producers (KPC + NDM). NG-Test CARBA 5^®^ showed a sensitivity of 98%, specificity of 100%, NPV of 87.5%, and PPV of 100%. All strains with NDM (17/17) and co-producers (KPC/NDM) were properly identified. The overall Kappa agreement coefficient between the PCR and NG-Test CARBA 5^®^ for Enterobacterales was 0.923 (CI 95% 0.661–1, *p* =< 0.001) (Table 1). Carbapenemase sequencing showed that *bla_NDM-1_* (16/51) and *bla_OXA-232_* (14/51) were the most prevalent (Table 2). NG-Test CARBA 5^®^ could not detect a bacterial strain harboring VIM carbapenemase (*bla_VIM-2_*). All *P. aeruginosa* carbapenemase producers (23/26) were MBLs; one was a co-producer of IMP and NDM carbapenemase (Table 2). The sensitivity and specificity of NG-Test CARBA 5^®^ were 82.6% and 100%, respectively, and 100% for PPV and 42.9% NPV. NG-Test CARBA 5^®^ was able to identify 1/1 NDM (100%), 5/8 IMP (62%), and 13/13 VIM (100%), but it was not able to identify the only co-producer (NDM + IMP). The Kappa agreement coefficient was 0.523 (CI 95% 0.145–0.901, *p* = 0.002) (Table 1). In *P. aeruginosa*, for carbapenemase sequencing, we mainly found *bla_VIM-2_* (13/26), three types of IMP (*bla_IMP-75_*, *bla_IMP-62_*, *bla_IMP-15_*), and only one *bla_NDM1_* (1/26).

## 3. Discussion

Carbapenem-resistant infections secondary to Enterobacterales or *P. aeruginosa* are of serious clinical worldwide concern, because the antimicrobial clinical treatment options are limited due to harbor resistance mechanisms inherent to many antibiotic classes [18]. Gram-negative bacteria are promiscuous microorganisms capable of sharing information mostly located in plasmids [19]. Epidemiological information has been kept relevant for many years in order to satisfy WHO recommendations regarding to the research of antimicrobial resistance [20]. Unfortunately, the Latin America region is one of the biggest stimulants for ß-lactamase production, with an enormous diversity of different antimicrobial mechanisms [21,22]. Rapid and accurate detection of carbapenemases is critical to diagnose and stop the spread inside the hospital and shorten the time of accurate treatment [23].

Compared to CLSI-endorsed routine methods for carbapenemases detection, NG-Test CARBA 5^®^ has advantages as it is a fast and simple test with minimal technical training. This facilitates its use in general hospitals or non-specialized clinical laboratories. The main advantage over colorimetric tests (RAPIDEC CARBA NP^®^, ß-CARBA^®^, CARBA NP) is its ability to detect enzymes with a weak hydrolytic capacity such as OXA-48-like, which results in multiple false negatives when employing colorimetric methods. On the other hand, an important advantage of NG-Test CARBA-5^®^ is the ability to discriminate between the different classes of carbapenemases [15,16,24,25]. There is also a trend of applying it directly into some clinical samples, such as positive blood culture bottles, in order to decrease results time [26,27,28].

Unlike colorimetric tests, the mCIM with and without EDTA has better performance for the detection of carbapenemases [12]. Unfortunately, it requires 24 h after microbiological isolation for its interpretation; as with CARBA NP, the discrepancy between those of the serine class and MBLs is only achieved with the method when modified by EDTA in order to reveal the MBLs. Finally, this method is not approved for use directly from positive blood culture bottles in comparison with NG-Test CARBA 5^®^ [11,29]. On the other hand, eCIM is only validated for use with Enterobacterales but not with *P. aeruginosa*; this is due to substantial discrepancies in the results [11].

Molecular methods based on polymerase chain reaction are usually the ideal methods for the identification of carbapenemases; however, they require technical training, and it is not available in several Latin American laboratories. Furthermore, the cost of homemade PCR, or even commercial kits, in lower-income countries exceeds the purchasing range, making it difficult to access these technologies in routine laboratories [30].

In our work, NG-Test CARBA 5^®^ exhibited excellent performance compared to PCR in Enterobacterales, with accuracy like phenotypic methods and achieved in less time. In Mexico, Garza Garcia et al. reported the distribution of carbapenem-encoding genes in Enterobacterales: the most frequently detected were *bla_NDM-1_* (81.5%) followed by *bla_OXA-48_* [17]. We found carbapenem-encoding genes such as *bla_NDM-1_*, *bla_NDM-5_*, *bla_KPC-2_*, *bla_KPC3_*, *bla_OXA-181_*, *bla_OXA-232_*, and *bla_VIM-2_*, but as we know *bla_KPC-82_* and *bla_VIM-67_* have not been previously described in Enterobacterales with NG-Test CARBA 5^®^ [31,32,33,34,35].

It is important to mention that NG-Test CARBA 5^®^ performance has been generally poor with *P. aeruginosa* [16]. This is relevant in settings such as ours, because close to 50% of clinical strains carbapanemase-producers *P. aeruginosa* are MBLs, predominantly VIM and IMP; on the other hand, according to some studies in Mexico, the second most frequent *P. aeruginosa* carbapenemase in Mexico belongs to GES, and this enzyme is not detectable with NG-Test CARBA 5^®^, which is a weakness of the test [17]. In our study, NG-Test CARBA 5^®^ was able to identify all VIM strains (13/13). However, in strains harboring IMP genes, NG-Test CARBA 5^®^ showed many inconsistencies (5/8 detected). This problem was already mentioned by Proton et al., who found a particular problem with IMP, but also found acceptable agreement with VIM and NDM [16]. We recognized three different types of IMP genes: *bla_IMP-15_*, *bla_IMP-75_*, and *bla_IMP-62_*; the first has already been described in the NG-Test CARBA 5^®^ literature, while the other two have not been previously described for NG-Test CARBA 5^®^ [16,36].

Compared to recent publications, this study had the advantage of evaluation employing the immunochromatographic test against multiple phenotypic tests while using molecular techniques as the gold standard [24,37]. Carbapenemases distribution was found to be like previously reported in Mexico, predominantly MBLs and OXA-48-like enzymes [17]. Although it is a weakness that we found few Enterobacterales VIM-producers and few *P. aeruginosa* NDM-producers, and no Enterobacterales IMP-producers, it is worth mentioning that sensitivity and specificity of phenotypic methods, especially of Carba NP-EDTA and eCIM, exhibited a worse performance in comparison to other publications. An opportunity of study is the sequencing of the carbapenemase encoding-genes, which might elucidate possible new enzymes not reported by this assay or characterize the carbapenemases that were falsely negative by the NG-Test CARBA-5^®^ [38,39].

MBLs are an emerging antimicrobial resistance threatening problem among Enterobacterales and *P. aeruginosa* strains present in nosocomial infections. Latin America has reported several hospital outbreaks related to different types of these enzymes, which is an increasing and perpetual problem [40,41,42]. It should be noted that even though OprD in *P. aeruginosa* is one of the main resistance mechanisms for carbapenems along with efflux pumps, this test might be used to identify the enzymatic mechanism and to help choose the adequate treatment. For carbapenem-resistant *P. aeruginosa*, we suggest incorporation of phenotypic assays or genome sequencing in order to elucidate the core mechanism according to local resistance; for example, one of the last CDC reports revealed a prevalence of 87% dysfunction of the porin OprD in *P. aeruginosa* [30,43].

This work shows the use of this assay in real life because we demonstrated that the NG-Test CARBA 5^®^ had excellent agreement (Kappa 0.9) for carbapenem resistance Enterobacterales (CRE); furthermore, in a region with a higher prevalence of MBLs and OXA-48 like carbapenemases, NG-Test CARBA 5^®^ could be an acceptable and fast option for routine laboratories. Based on our findings, we propose an algorithm (Figure 2) for CRE detection, applicable in countries with a high prevalence of antimicrobial resistance with an increased burden of ß-lactamases [17].

In conclusion, NG-Test CARBA 5^®^ is a fast and easy-to-use assay. In Enterobacterales, we found excellent agreement with molecular tests (Kappa 0.9). On the other hand, despite the low agreement in *P. aeruginosa* (Kapa 0.5), we suggest that this test might be used as a tool for identifying the enzymatic mechanisms with the aid of other assays.

## 4. Materials and Methods

**Clinical strains.** We included all non-duplicated Enterobacterales and *P. aeruginosa* bacteria showing resistance to at least one carbapenem drug from January 2019 to March 2020. We defined a non-duplicated strain as the first isolated bacterial found in a patient clinical sample independent of isolation sites. There were four participating hospitals: Instituto Nacional de Rehabilitación Luis Guillermo Ibarra Ibarra (INRLGII), Instituto Nacional de Cancerología (INCan), Hospital General Dr. Manuel Gea González (HGDMGG), and Hospital Civil de Guadalajara Fray Antonio Alcalde (HCGFAA). All centers performed the identification and susceptibility tests using the Vitek 2 System (BioMérieux, Marcy 1′Étoile, France).

**Minimal inhibitory concentrations**. Broth microdilution was employed to corroborate carbapenem resistance for all strains following the CLSI recommendations [18]. The evaluated antibiotics were: ertapenem (not in *P. aeruginosa*), meropenem, imipenem, and doripenem. Breakpoints were defined according to the M100 from CLSI [11].

**Phenotypic assays for the presence of carbapenemases.** All assays were performed in the Clinical Microbiology Laboratory at INR LGII. Phenotypic assays were performed according to CLSI M100 for mCIM, eCIM, CARBA NP, and CARBA NP-EDTA. In the case of Enterobacterales, the eCIM and CARBA NP-EDTA tests were performed to identify MBLs; only the eCIM test was used to identify *P. aeruginosa* isolates [11].

**Carbapanemase related-genes detection**. Bacteria DNA was extracted by thermal shock (95 °C/20 min) in PBS. Tubes were centrifuged and supernatants were separated in new tubes. PCR conditions were the following: 10X PCR Buffer (New England BioLabs, Ipswich, MA, USA) containing 2 mM of MgCl_2_, 0.2 mM of each of the dNTPs (Invitrogen, Waltham, MA, USA), 10 pmol of each oligonucleotide (Appendix A), 1.5 U of Taq polymerase (BioLabs, USA), and 5 μL of DNA; PCR was performed according to manufacturer recommendations. The amplification was carried out using the following program: 1 cycle of 5 min at 95 °C, 35 cycles of 50 s at 95 °C, 60 s at 55 °C, and 50 s at 68 °C, followed by 1 final cycle of 10 min at 68 °C (Veriti, Applied Biosystems, Waltham, MA, USA). PCR products were separated in a 1% agarose gel at 100 volts for 1 h and SYBR Green (Invitrogen, Waltham, MA, USA) was used such as DNA intercalant. Bands were visualized in GelDocTM XR+ with Image LabTM Software (BioRad, Hercules, CA, USA).

**Sequencing**. Prior to the sequencing process, re-amplification and labeling using the BigDye^®^ Direct Cycle Sequencing Kit (Applied Biosystems, Waltham, MA, USA) were performed. This step was done separately for each one of the primers previously described for the amplification; thus, sequencing was done using the forward and the reverse primers separately POP-7 (Applied biosystems, EUA). GenBank was used to compare the sequences obtained with those of reference organisms in this database. The identification was corroborated by performing independent sequencing for each primer.

**NG-Test CARBA 5^®^.** The test was performed according to manufacturer recommendations. The lecture was executed after 15 min. The validated reaction was taken when a red line appeared in position C. The type of carbapenemase was defined when the red line appeared in the respective letter (K: KPC, O: OXA-48-like, V: VIM, I: IMP, and N for NDM).

**Statistical analysis.** Measures of central tendency were calculated. The detection capacity was calculated with the Kappa coefficient between PCR and phenotypic assays, as well as with the Kappa coefficient between PCR and NG-Test CARBA 5^®^. Comparable agreements were defined as those closer to 1, statistical significance was defined as *p* ≤ 0.05. Sensitivity, specificity, negative predictive, and positive predictive values were also calculated for phenotypic assays and NG-Test CARBA 5^®^ for Enterobacterales and *P. aeruginosa*. Software SPSS 22 was used for statistical analysis.

## Figures and Tables

**Figure 1 antibiotics-12-00096-f001:**
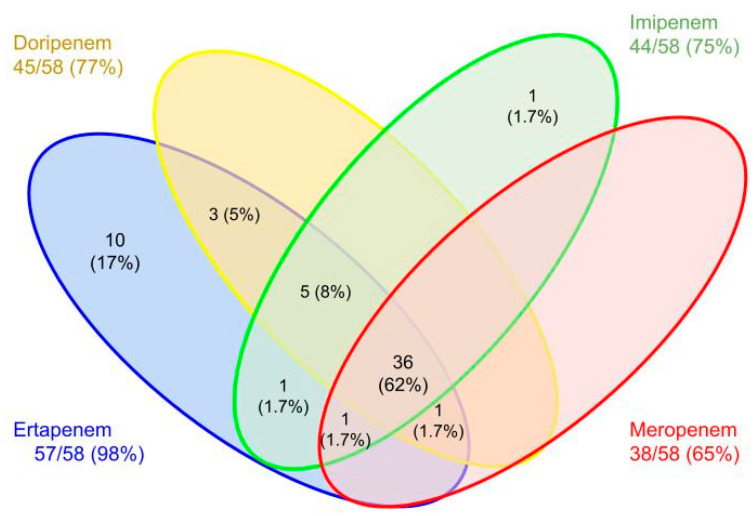
Carbapenem resistance distribution in Enterobacterales.

**Figure 2 antibiotics-12-00096-f002:**
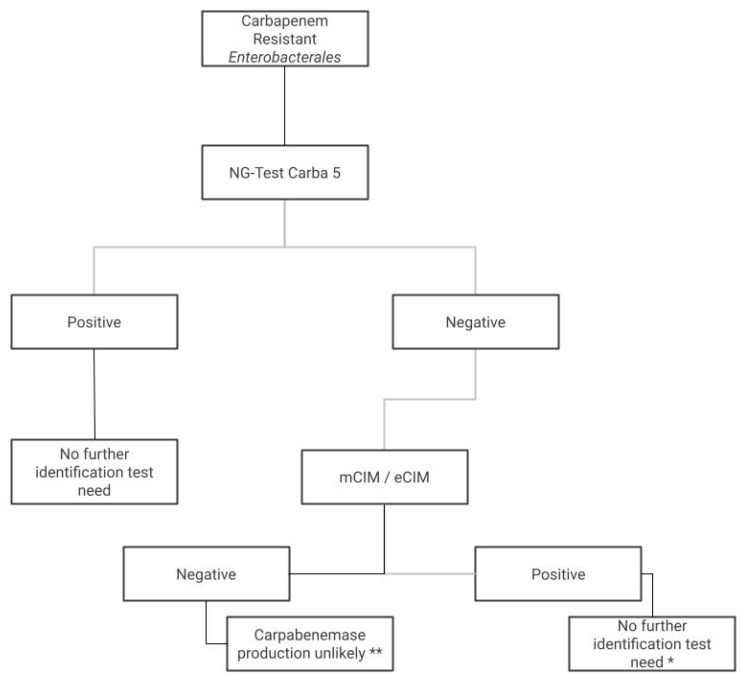
Algorithm for CRE detection. * We don’t recommend further testing in a routine laboratory that does not have PCR assays or genome sequencing; instead, we recommend contacting the central state laboratory. ** The mCIM test has a NPV close to 100%; in case of a negative test, the carbapenemase production seems to be unlikely and these strains must be investigated for another carbapenem-resistance mechanism like hyper AMP-c production.

**Table 1 antibiotics-12-00096-t001:** Accuracy of the phenotypic test and NG-Test CARBA 5^®^ compared to PCR.

Microorganisms	n	Test	Sensitivity	Specificity	PPV	NPV	Kappa (CI 95%)	
Enterobacterales	58	mCIM	100	100	100	100	1 (1–1)	*p* < 0.001
		mCIM/eCIM	86	100	100	50	0.603 (0.27–0.85)	*p* < 0.001
		CARBA NP	67	100	100	30	0.347 (0.15–0.55)	*p* < 0.001
		CARBA NP +EDTA	63	100	100	27	0.29 (0.1–0.48)	*p* < 0.001
		CARBA 5	98	100	100	87	0.923 (0.661–1)	*p* < 0.001
*P. aeruginosa*	26	mCIM	100	100	100	100	1 (1–1)	*p* < 0.001
		mCIM/eCIM	83	100	100	43	0.523 (0.001–0.9)	*p* = 0.002
		CARBA NP	61	100	100	25	0.264 (0.001–0.58)	*p* = 0.047
		CARBA 5	82.6	100	100	42.9	0.523 (0.14–0.9)	*p* = 0.002

Numbers expressed in percentage (%). PPV: Positive Predictive Value; NPV: Negative Predictive Value; mCIM: modified carbapenem inactivation method; eCIM: modified carbapenem inactivation method with EDTA.

**Table 2 antibiotics-12-00096-t002:** Performance of phenotypic assays, NG-Test CARBA 5^®^, and PCR for Enterobacterales and *P. aeruginosa*.

Microorganisms		Genomic Sequencing	mCIM/eCIM		Carba NP/EDTA *		PCR		Carba 5^®^	
Enterobacterales	N		Detected	Non Detected	Detected	Non Detected	Detected	Non Detected	Detected	Non Detected
KPC (10)	8	KPC-2	8	0	8	0	8	0	8	0
	1	KPC-82	1	0	1	0	1	0	1	0
	1	KPC-3	1	0	1	0	1	0	1	0
NDM (17)	16	NDM-1	12	4	16	0	16	0	16	0
	1	NDM-5	1	0	1	0	1	0	1	0
VIM (3)	2	VIM-2	1	1	1	1	2	0	1	1
	1	VIM-67	1	0	0	1	1	0	1	0
OXA-48 like (19)	14	OXA-232	14	0	3	11	14	0	14	0
	3	OXA-48	3	0	1	2	3	0	3	0
	2	OXA-181	2	0	0	2	2	0	2	0
Co-producers	2	NDM-1 KPC-2	0	2	0	2	2	0	2	0
No-carbapenemases producers	7	-	0	7	0	7	0	7	0	7
Total	58		44	14	32	26	51	7	50	8
** *P. aeruginosa* **	N		Detected	Non detected	Detected	Non detected	Detected	Non detected	Detected	Non detected
IMP (8)	4	IMP-75	2	2	4	0	4	0	1	3
	3	IMP-62	3	0	2	1	3	0	3	0
	1	IMP-15	1	0	1	0	1	0	1	0
VIM	13	VIM-2	11	2	5	8	13	0	13	0
NDM	1	NDM-1	1	0	1	0	1	0	1	0
Co-producers	1	IMP-75 + NDM-1	1	0	1	0	1	0	0	1
No-carbapenemases producers	3		0	3	0	3	0	3	0	3
Total	26		19	7	14	12	23	3	19	7

* Carba NP-EDTA was not performed for *P. aeruginosa*. mCIM: modified carbapenem inactivation method; eCIM: modified carbapenem inactivation method with EDTA; PCR: polymerase chain reaction.

## Data Availability

Data is contained within the article or Appendix A.

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
