# Peer review of "Comparison of Lateral Flow Immunochromatography and Phenotypic Assays to PCR for the Detection of Carbapenemase-Producing Gram-Negative Bacteria, a Multicenter Experience in Mexico"

_antibiotics, 2023, doi:10.3390/antibiotics12010096_

Round 1
Reviewer 1 Report
Dear Authors:
Although the manuscript is of interest and data are quite well presented (even with the disclosed limitations), these analyses suffer from some shortcomings and a lack of relevant data.
Major revision comments:
Line 83: How did the authors confirm the collected strains were non-duplicated?
Line 236: Please provide specimen descriptions in a supplementary table including the list of strain identifers (IDs), isolation sites (stool, blood, urine, etc), collection dates (month/year), antimicrobial profile, and identified resistance genes. These data are important. Please check the Instructions for Authors: https://www.mdpi.com/journal/antibiotics/instructions
Minor revision comments:
Please consider changing the title of the article on:
The comparison of NG-Test CARBA 5® based on lateral flow immunochromatography and other phenotypic assays to PCR for the detection of carbapenemase-producing Gram-negative bacteria, a multicenter experience in 3 Mexico.
Affiliation:
Line 5 and throughout the article: check for redundant spaces
Line 14: "Luis Guillermo Ibarra Ibarra," – whether quotation marks are needed?
Abstract:
Line 21, 24, 28, 33: Please check the abstract writing rules on: https://www.mdpi.com/journal/antibiotics/instructions
Line 21 and throughout the article: Enterobacterales should be written without italics.
Line 21: P. aeruginosa – please provide the full name of the genus and species for the first time.
Line 30: P. aeruginosa – should be 23 instead of 24. Please also check the percentage.
Line 31 and throughout the article: Please replace sensibility on sensitivity.
Key words: Species names should be in italics. Please check the whole article.
Introduction:
Line 48: Please replace the comma with a period (5.3%).
Line 81: Put a period after MBLs please.
Results:
Line 83 and 84: 58/84~ 69.0%; 26/84~31.0%. Please check the percentage.
Line 106: could not detect a bacterial strain harboring VIM carbapenemase… Please replace with: could not detect the one bacterial isolate harboring VIM carbapenemase
Table 2: Please add the row “Total” as for Enterobacterales
Table 2: Should be OXA-48 like instead of OXA-48
Figure 2: Explain what * and ** mean.
Figure 2: Why “No further identification test need”?
Discussion:
Line 161, 184: Should be OXA-48 like instead of OXA-48.
Line 165: such as blood culture bottles…. Please replace with: such as positive blood culture bottles
Line 167: Please enter the abbreviation CIM.
Line 223: Please define CRE and other abbreviations when they appear for the first time in the abstract, main text, and the first figure or table caption.
Material and methods:
Line 241 and throughout the section: Please state the name of the manufacturer, city, and country from where the equipment was sourced.
Line 255: Please enter Supplementary Table S1.
Supplementary materials:
Line 283: Please check Carbapenemases.
Table S1: Please enter the correct form of writing the names of genes.
Line 284: It should be: …..the proportion of species of the Enterobacterales.
Author Response
Dear reviewer.
We appreciate the thoughtful revision comments you have made. We approve most of the recommendations.
We acknowledge the major and minor revision comments you made, so this is the answer to every question
Major revisions:
- Line 83: How about non-duplicated stains? Thanks for your comment, we add an explaining line 258 and 259 in the new document.
- We defined a non-duplicated strain as the first isolated bacterial found in a patient clinical sample independent of isolation sites
- Supplementary table with specimen descriptions. Thanks for the idea we add a new table (Supplementary table 3) with a sort of variables like strain ID, collection hospital, specie, isolation sites (blood, urine, respiratory sample…), carbapenems MIC´s, and genome sequencing.
Minor revisions:
- Affiliation, abstract, key words, and discussion, supplementary section we approve all the recommendations. Thanks for that
- We appreciate the change in the title, but we want to keep it on our way. Thanks for the comment
- Results
- We amend the percentages in Line 86 and 87, thank you.
- 69% and 31%
- The recommendation in line 106, for replace with: could not detect the one bacterial isolate harboring VIM carbapenemase. We appreciate de comment, but we had to deny the recommendation because there are more than one VIM carbapenemase, that’s not the only one. Thank you.
- We add de total file in Table 2 for Enterobacterales and change de OXA-48 for OXA-48 like in the microorganism row. Thank you
- We add an explanation for de * and ** in the figure. Thank you
- We amend the percentages in Line 86 and 87, thank you.
- Material and methods:
- We add in an upper line de supplementary table 1. Thank you.
Sincerely Thanks
The Authors.

Reviewer 2 Report
Line 266- Which sequencing platform was used.
Line 225- If NG test CARBA 5 needs sequencing to be done, how this could be fast test for routine laboratories. Can you elaborate on time and cost involved per test for NG test CARBA 5.
Line 230- In clinical settings, how NG test- CARBA 5 result would change the management.
Line 232- What modification can be done to improve Kappa value for P. aeruginosa.
Author Response
Dear reviewer
We appreciate the thoughtful revision comments you have made. We answer made a list of answers to your recommendations.
- Line 266- Which sequencing platform was used.
- Thanks for noticing, we use a POP-7 (Applied biosystems, EUA), we add in the line 292
- Line 225. If NG test CARBA 5 needs sequencing to be done, how this could be fast test for routine laboratories. Can you elaborate on time and cost involved per test for NG test CARBA 5.
- Thaks for the recommendation and thoughts. The purpose of the study was to evaluate the diagnostic accuracy of NG-Test CARBA 5, therefore we compared whit a gold standard assay (polymerase chain reaction), and with phenotypical assays, also we add the sequencing to improve our enzyme genotyping. The NG-TEST Carba 5 needs minimal equipment, we performed the assay with what’s is included in the manufactures kit. In Mexico a very few hospitals or laboratories has the equipment to perform a sequencing or a polymerase chain reaction. We believe that with our results, we can recommend the use of NG-TEST Carba 5 in routine laboratories that lack of polymerase chain reaction or sequencing. In Mexico the exactly cost of polymerase chain reaction and sequencing its variable, but it is nor affordable for the majority our hospitals and routine laboratories. We believe that an economic scope can answer the question properly.
- Line 230. In clinical settings, how NG test- CARBA 5 result would change the management?
- Thanks for the comment. The IDSA society guidelines for the treatment of multidrug resistance Gram negative-rods encourages the clarification of the resistance mechanism; an example are carbapenemases microorganisms producers Class A or OXA-48 like could be treated with new antibiotics like Ceftazidime-Avibactam or Meropenem-Varbobactam; unlike the Class B, that should not be treated by these antibiotics. Therefore, the rapid identification of the carbapenemases could improve the antibiotic use and perhaps the clinical outcome. However, the objective answer to that question requires a clinical trial, that is not the purpose of our work
- Line 232- What modification can be done to improve Kappa value for P. aeruginosa.
- Thanks for the comment. As we see in the work of Volland et al, they improve the detection of IMP variants in an enhanced NG-TEST CARBA 5, like a 2.0 version. We believe that the same technique can replicate with a new pool of IMP variants undetected
Sincerely Thanks
The Authors.
